# Transcriptomics Reveal Molecular Differences in Equine Oocytes Vitrified before and after In Vitro Maturation

**DOI:** 10.3390/ijms24086915

**Published:** 2023-04-07

**Authors:** Daniel Angel-Velez, Tim Meese, Mohamed Hedia, Andrea Fernandez-Montoro, Tine De Coster, Osvaldo Bogado Pascottini, Filip Van Nieuwerburgh, Jan Govaere, Ann Van Soom, Krishna Pavani, Katrien Smits

**Affiliations:** 1Department of Internal Medicine, Reproduction and Population Medicine, Faculty of Veterinary Medicine, Ghent University, Salisburylaan 133, 9820 Merelbeke, Belgium; 2Research Group in Animal Sciences—INCA-CES, Universidad CES, Medellin 050021, Colombia; 3Laboratory for Pharmaceutical Biotechnology, Faculty of Pharmaceutical Science, Ghent University, 9000 Ghent, Belgium; 4Department of Theriogenology, Faculty of Veterinary Medicine, Cairo University, Giza 12211, Egypt; 5Department for Reproductive Medicine, Ghent University Hospital, Corneel Heymanslaan 10, 9000 Gent, Belgium

**Keywords:** cryopreservation, horse oocyte, RNA sequencing

## Abstract

In the last decade, in vitro embryo production in horses has become an established clinical practice, but blastocyst rates from vitrified equine oocytes remain low. Cryopreservation impairs the oocyte developmental potential, which may be reflected in the messenger RNA (mRNA) profile. Therefore, this study aimed to compare the transcriptome profiles of metaphase II equine oocytes vitrified before and after in vitro maturation. To do so, three groups were analyzed with RNA sequencing: (1) fresh in vitro matured oocytes as a control (FR), (2) oocytes vitrified after in vitro maturation (VMAT), and (3) oocytes vitrified immature, warmed, and in vitro matured (VIM). In comparison with fresh oocytes, VIM resulted in 46 differentially expressed (DE) genes (14 upregulated and 32 downregulated), while VMAT showed 36 DE genes (18 in each category). A comparison of VIM vs. VMAT resulted in 44 DE genes (20 upregulated and 24 downregulated). Pathway analyses highlighted cytoskeleton, spindle formation, and calcium and cation ion transport and homeostasis as the main affected pathways in vitrified oocytes. The vitrification of in vitro matured oocytes presented subtle advantages in terms of the mRNA profile over the vitrification of immature oocytes. Therefore, this study provides a new perspective for understanding the impact of vitrification on equine oocytes and can be the basis for further improvements in the efficiency of equine oocyte vitrification.

## 1. Introduction

Effective equine oocyte cryopreservation has great potential for basic research and commercial applications [1]. Healthy foals have been produced from vitrified-warmed oocytes in horses [2,3], but the formation of blastocysts and the production of live offspring from vitrified and warmed oocytes is not yet efficient [4,5,6,7,8]. Oocytes can be vitrified either at the immature or mature stage. In humans, oocyte vitrification is routinely performed with in vivo matured oocytes, collected after hormone stimulation, resulting in similar clinical outcomes between fresh and vitrified oocytes [9,10]. However, if vitrification is performed before [3,7,11,12,13,14] or after in vitro maturation [6,15,16,17,18], the developmental capacity of oocytes in both humans and domestic animals is affected. Whether vitrification should preferentially be performed before or after in vitro maturation remains controversial and is potentially influenced by species-specific differences. In cattle, a higher cleavage rate was achieved after the vitrification of in vitro matured oocytes compared to vitrified immature oocytes [19,20], but blastocysts and calves have been obtained after the vitrification of both immature and mature oocytes [21,22,23,24,25]. In pigs, immature oocytes are also more sensitive to cryopreservation than oocytes in the metaphase II (MII) stage [26]. However, in goats, oocytes in the germinal vesicle (GV) stage enclosed by compact cumulus cells were more tolerant to cryopreservation than oocytes vitrified after in vitro maturation with and without cumulus cells [27].

In the horse, using in vivo matured oocytes for oocyte vitrification leads to the best developmental outcome (40% blastocyst rate) [28]. However, the application potential of this method is limited because superovulation is ineffective in horses [2,29]. As a consequence, most research on equine oocyte cryopreservation has been performed with immature oocytes, but the results have been poor [4,5,6,7,30]. Only a few studies have evaluated the vitrification outcome after in vitro maturation, but the results have not been satisfactory either [6,31]. Therefore, in the horse, the ideal meiotic stage for cryopreservation remains to be determined.

Low embryo development rates from vitrified-warmed oocytes may be associated with damage to several oocyte components, including the cytoskeleton, organelles, and cell membrane [8,32,33,34,35]. When oocytes are vitrified in MII, their spindles are sensitive to low temperatures and cryoprotective agents [36,37,38,39]. Theoretically, this damage can be avoided by vitrification of GV stage oocytes, as the spindle apparatus has not yet been formed at this stage [40]. However, oocytes vitrified at the immature stage must complete meiosis after warming, which may further influence oocyte competence. Many approaches, such as the use of antioxidants [8], ice-blockers [41], different mixtures of cryoprotectants [7,42], and an increased cooling rate [3] have been tried to optimize oocyte vitrification, but the achieved outcomes are still unsatisfactory. Different studies have reported differential expression in development-related genes after vitrification [43,44,45,46] and the transcriptomes of vitrified vs. fresh oocytes have been reported in humans [47,48,49], mice [50,51], cattle [52,53,54], and porcine [55]. To the best of our knowledge, no transcriptomics studies have been performed on the vitrified oocytes of horses.

The transition from oocyte via zygote until genome activation is primarily dependent on maternal RNAs and proteins that have been accumulated and stored in the cytoplasm during oocyte growth, a molecular process known as “molecular maturation” [56,57,58,59,60]. Before meiotic resumption, fully grown GV oocytes are transcriptionally silent until zygotic genome activation occurs after fertilization [61,62]. However, meiotic maturation and the oocyte-to-embryo transition require temporal changes in polyadenylation levels and selective degradation to maintain developmental competence [63,64,65,66]. Some of these molecular adaptation variations are related to oocyte quality [65,67]. Therefore, obtaining a better understanding of the molecular effects of oocyte vitrification would guide the modification of oocyte vitrification protocols to improve their efficiency. Unraveling the differences between fresh and vitrified oocytes before or after in vitro maturation might further help to elucidate which meiotic stage, GV or MII, is more suitable for vitrification.

The development of RNA sequencing (RNA-seq) enables the genome-wide analysis of the transcriptome in tissues or cells and is primarily used to quantify differences in gene expression related to biological processes [68]. RNA-seq can help to understand the effect of vitrification on crucial oocyte components and pathways. Here, we aimed to analyze the differences in the transcriptomes of fresh in vitro matured oocytes and in vitro matured oocytes vitrified before and after in vitro maturation. Our results contribute to the understanding of the mRNA profiles of oocytes undergoing vitrification at different meiotic stages, highlighting the cytoskeleton, spindle formation, calcium and cation ion transport, and homeostasis as the main affected pathways and indicating a potential advantage at the molecular level of vitrification after in vitro maturation.

## 2. Results

### 2.1. Vitrification Impairs Oocyte Survival and Maturation

We compared the transcriptome profile from three groups of equine in vitro matured oocytes: (1) fresh in vitro matured oocytes as a control (FR), (2) oocytes vitrified after in vitro maturation (VMAT), and (3) oocytes vitrified immature, warmed, and in vitro matured (VIM) (Figure 1). Survival and maturation rates were compared for all groups. FR and VMAT oocytes underwent in vitro maturation together, resulting in significantly higher maturation rates, compared to VIM (*p* = 0.0004; Figure 2, Appendix A). The survival rate (i.e., intact oolema after warming) was higher for VIM than for VMAT (*p* = 0.0004; Figure 2, Appendix A). However, as oolema assessment is difficult in immature cumulus-oocyte complexes (COCs) due to the presence of cumulus cells, and degeneration may not be perceived clearly, a final efficiency (oocytes stored for transcriptomics/initial number of oocytes) was compared. The final efficiency for VIM was lower than for FR (*p* = 0.0007) and similar to that of VMAT (*p* = 0.1879). The efficiency of VMAT was not significantly different from that of FR (*p* = 0.0954; Figure 2, Appendix A).

### 2.2. Vitrification of Immature and Mature Oocytes Affects the mRNA Profile

On average, 23 oocytes were pooled per replicate. Each sample contained an average of 31 million reads, with a standard deviation of 22 million reads. On average, 88% of the reads were uniquely mapped to the reference genome (Equus caballus (EquCab3.0, release 108)) by STAR 2.7.10b. Differential expression categorization (i.e., up or downregulation) is associated with the first factor in the comparison. A total of 112 genes were differentially expressed (DE) in all three comparisons (VIM vs. FR; VMAT vs. FR; and VIM vs. VMAT). Vitrification modified the transcriptome since 78 genes were DE between vitrified (VIM and VMAT) and fresh (FR) oocytes. We identified 46 DE genes (14 upregulated and 32 downregulated) in VIM compared with FR (Appendix A, Figure 3) and 36 genes were DE in VMAT (18 upregulated and 18 downregulated) compared to FR (Appendix A, Figure 3). Finally, a comparison between the two vitrified groups (VIM vs. VMAT) resulted in 44 DE genes (20 upregulated and 24 downregulated) (Appendix A, Figure 3). Common DE genes among the groups are summarized in Figure 3A. We identified four DE genes for both vitrified groups compared to the fresh control oocytes. Three of them are downregulated in both vitrified groups (G3BP2: G3BP Stress Granule Assembly Factor 2, TOMM70: translocase of outer mitochondrial membrane 70, and RCAN3: RCAN family member 3) and one is upregulated (COMMD10: COMM Domain Containing 10). Similarly, four DE genes are common among VIM vs. VMAT and VMAT vs. FR. Two of those, REEP3: receptor accessory protein 3 and CALCOCO2: calcium binding and coiled-coil domain 2, were upregulated in both comparisons, while TRAF6: TNF receptor-associated factor 6 and DOCK8: dedicator of cytokinesis 8 were downregulated. Finally, in the comparisons of VIM vs. VMAT and VIM vs. FR, six DE genes (STK24: serine/threonine kinase 24, REEP6: receptor accessory protein 6, BFAR: bifunctional apoptosis regulator, CEP20: bifunctional apoptosis regulator, ZNF638: zinc finger protein 638, and a novel gene ENSECAG00000017318) were common and were all downregulated in VIM vs. FR and upregulated in VIM vs. VMAT.

### 2.3. Vitrification Affects Pathways Involved in Oocyte Development

The SimplifyEnrichment analysis of VIM vs. FR identified that the predominant pathways affected in biological functions were cytoskeleton organization and assembly, mitogen-activated protein kinase (MAPK) unfolded cascade stimulus, and calcium ion transport and signaling (Figure 4A), cilium projection was the most important as a cellular component (Appendix A), and binding in molecular function (Appendix A). A comparison of VMAT vs. FR revealed GO clusters enriched in ubiquitination, calcium and cation ion transport and homeostasis, spindle formation for biological function (Figure 4B), spindle and nuclear complex for cellular component (Appendix A), and cation ion transporter for molecular function (Appendix A). Finally, the comparison between both vitrified groups (VIM vs. VMAT) indicated altered pathways involved in calcium and cation ion transport and homeostasis, actin depolymerization, transcription, and cell cycle regulation for biological function (Figure 4C), the mitochondria and DNA polymerase complex for the cellular component (Appendix A) and exonuclease, ion transport, and kinase activity in molecular function (Appendix A).

## 3. Discussion

Oocyte cryopreservation reduces the potential for embryonic development in mammals [7,69,70,71], and although different approaches have attempted to improve vitrification efficiency [3,5,7,8,41,42], we still lack a comprehensive understanding of the effect of vitrification on the equine oocyte. This study uses the advantages of RNA sequencing to provide novel insights into the impact of vitrification on the equine oocyte and in the molecular variations that occur depending on whether vitrification occurs at the immature or mature stage.

It is well known that de novo mRNA synthesis ceases with the growth of the oocyte and that fully grown oocytes are transcriptionally silent [56,57,58,59,60,62]. However, in vitro maturation [65,72], or specific treatments such as vitrification, can be associated with modifications in the mRNA profile [51,52,53,55]. These alterations are linked to the degradation and polyadenylation of maternal RNA, which influence gene and protein expression [65,73]. The developmental competence of the oocyte depends on a well-orchestrated combination of storage and degradation of RNA, in which both over- or under-degradation may affect embryonic development [63,64]. Oocyte vitrification has been mainly linked with the downregulation of mRNAs [48,50,52,53], which might be correlated with the decrease in stored mRNA found in human oocytes after vitrification [74]. This is in line with our results for the VIM group, but not for VMAT, in which an equal number of mRNAs were up- and downregulated, compared to the fresh control group. Importantly, the “up- or downregulation” of mRNAs does not necessarily reflect the functional effect of these mRNAs. It may be that the “downregulation” of mRNA, representing the observation of a lower number of copies of this mRNA at that moment in that treatment group, is actually caused by increased translation to the corresponding protein.

According to our findings, the mRNA profile of equine oocytes was altered by vitrification, with a total of 78 DE genes when comparing all vitrified oocytes with the fresh control group. Comparisons in humans and mice between fresh and vitrified in vivo MII oocytes resulted in pronounced discrepancies between studies. Whereas a first study in mice [51] showed minor or no changes in the transcriptome, a recent study reported 4747 DE genes [50]. Similarly, in humans, Monzo et al. (2012) [48] found 608 DE genes using GeneChip arrays, and later, Huo et al. (2021) [75] and Barberet et al. (2022) [49] found notable differences, with a total of 1987 and 108 DE genes, respectively, using single-cell RNA-seq. In pigs and cattle, comparisons were made between fresh in vitro matured and oocytes vitrified at the GV stage followed by in vitro maturation. While 37 genes were DE in swine [55], in cattle, the DE genes varied from 52 [52] up to 609 [54]. Remarkably, the latter study by Zhang et al. (2020) [54] demonstrated the importance of the vitrification protocol, showing that only changing the temperature and the concentration of cryoprotectants dramatically altered the gene expression profiles, ranging from 218 to 609 DE genes for the different treatments, indicating the potential to reduce the detrimental impact of the vitrification procedure by protocol optimization. Comparing different studies of the same species only revealed an alteration in a limited number of common genes. In humans, only seven DE genes were common in studies by Huo et al. (2021) [75] and Barberet et al. (2022) [49]. Additionally, they displayed an opposite variation of expression. In mice, only two DE genes were common, also with an opposite expression [50,51], and in cattle, no DE genes were common between the studies by Huang et al. (2018) [52], Wang et al. (2017) [53], and the 21 DE genes specified by Zhang et al. (2020) [54]. These discrepancies could be associated with the vitrification method, gene analysis technique (i.e., GeneChip arrays vs. RNA-seq assessment), and the number of oocytes analyzed (i.e., single cell vs. pool).

One of the most important factors that influence the vitrification outcome is the meiotic stage in which the oocyte is vitrified [1]. In our study, vitrification after in vitro maturation resulted in slightly less modification of the transcriptome: 36 DE genes were found in VMAT vs. FR while VIM vs. FR resulted in 46. This is in contrast with bovine studies where a more similar mRNA profile was found when vitrification was performed before instead of after in vitro maturation, with 53 DE genes [52] and 102 DE genes [53], respectively. However, those studies used different vitrification protocols which can affect the outcome [54]. Only Gao et al. (2017) [51] directly compared the effect of vitrification at the mature vs. immature stage in mice. In this study, vitrification after in vivo maturation did not alter the transcriptome of the oocyte compared to fresh in vivo matured oocytes, while vitrification at the GV stage followed by in vitro maturation resulted in 18 DE genes compared to the fresh in vivo matured oocytes. The similarity in the transcriptomic profile among fresh and vitrified in vivo matured oocytes coincides with the favorable developmental results of the in vivo matured oocytes used for vitrification in humans [76] and horses [2,28]. Unfortunately, Gao et al. (2017) did not compare oocytes vitrified before and after in vitro maturation. In our study, the comparison between VMAT and VIM showed that vitrification at the MII stage may be preferential in horse oocytes, VIM is associated with the downregulation of developmentally important genes, including *TRAF6*: TNF receptor-associated factor 6, DOCK8: dedicator of cytokinesis 8, and LAMTOR1: late endosomal/lysosomal adaptor; MAPK and MTOR activator 1 Involvement in embryo development are documented by the association of TRAF6 with NF-κB-mediated transcription and apoptosis [77,78,79], while DOCK8 is crucial for cell migration during embryo development [80,81] and LAMTOR1 regulates several cellular processes including the organization and remodeling of cytoskeletal dynamics, regulation of cell cycle progression, cell polarity and migration, gene expression, and cell survival [82,83].

Vitrification alters the mRNA profile, but it seems that no common transcriptomic signature associated with vitrification has been observed across studies, including ours. Yet, pathway enrichment analysis does indicate common pathways which are similarly affected in different studies across species. In our study, pathways related to the cytoskeleton and spindle formation, calcium transport, and homeostasis were altered in both vitrified groups. Vitrification at the GV stage additionally changed the mitogen-activated protein kinase (MAPK) pathway, which was also affected in vitrified mouse MII oocytes [50], and the vitrification of mature oocytes further revealed a disturbance of the ubiquitination pathway, as observed after MII vitrification in humans [48]. All the altered pathways in our study are directly related to the cell cycle, meiosis, and cell division [84,85,86], which are the main pathways distorted in vitrified oocytes in humans [75] and cows [52,53]. Other pathways related to mRNA and RNA catabolic processes and membrane-bound and intracellular organelles were affected after the vitrification of human [49] and bovine oocytes [53], which coincide with the clusters found in VIM vs. VMAT, where the metabolic process and transcription initiation in the biological function and the polymerase complex and organelle membrane in the cell component were affected. Remarkably, pathways related to calcium and ion transport, which were common in our three comparisons, have not been mentioned in other studies, except in cumulus cells after vitrification in porcine oocytes [55]. However, in cows, Ca^2+^ regulation genes, such as *CALM:* calmodulin, *SSRG:* Signal Sequence Receptor, Gamma, and *GPC5D:* G-Protein Coupled Receptor Family C Group 5 Member D, were downregulated in vitrified oocytes [53]. Regarding the pathways affected in the comparison of meiotic stages, important pathways required for acquiring developmental competence such as phosphorylation kinase, ion cation transport, and homeostasis as well as the mitotic checkpoint, mitochondrial inner membrane, and exonuclease activities were enriched in VIM vs. VMAT.

The similar efficiency in terms of the survival and maturation of VMAT and FR, the more disturbed transcriptome in VIM, and the favorable development and transcriptome profiles published after vitrification of in vivo matured oocytes suggest that vitrification after in vitro maturation may be preferential and that the optimization of in vivo-like maturation systems could be a key factor to further improve the results. Likewise, strategies to optimize oocyte vitrification could target the pathways mentioned above. Regarding horses, some research has been performed to improve the mitochondrial function with melatonin [8], but other approaches applied in other species, such as spindle-stabilizing molecules, i.e., cytochalasin B, paclitaxel, and docetaxel, in the culture and vitrification or warming media [87,88,89], or modifying calcium levels in the vitrification medium [90,91], may be promising alternatives for horse oocytes too.

Although this study describes for the first time the influence of vitrification on the mRNA profile of in vitro matured oocytes, showing that our findings are highly correlated with the known structural damage after vitrification, there are some limitations. Firstly, it has been demonstrated that the vitrification protocol affects the mRNA profile of the oocyte [54], and in our study, only one vitrification protocol was tested. Secondly, although oocytes are transcriptionally silent during in vitro maturation, the adenylation and degradation of maternal mRNA is a dynamic process [73,92], and we only evaluated the mRNA profile at a specific time-point (i.e., after 30 h of in vitro maturation). Therefore, some variation might be associated with the maturation process. Thirdly, all oocytes were collected post-mortem, and differences in developmental potential have been reported between oocytes derived from slaughterhouse ovaries vs. those collected by ovum pick-up in live animals [93]. Hence, these differences can also be reflected in the transcriptome and may be considered in further studies. Finally, further validation of our RNA-sequencing results by targeting specific genes or by functional validation will improve the biological relevance of our findings.

In conclusion, our results show that (1) vitrification before or after in vitro maturation influences the mRNA profile of equine oocytes in a different way, (2) pathways related to the cytoskeleton, spindle formation, calcium transport, and homeostasis are commonly affected in vitrified equine oocytes, and (3) vitrification after in vitro maturation provides subtle advantages in terms of the mRNA profile of developmentally important genes. However, to corroborate which meiotic stage is more suitable for vitrification, developmental experiments should be evaluated. This study contributes to the understanding of modifications in the mRNA profile in vitrified and in vitro matured oocytes and offers a basis for further improvements to the efficiency of equine oocyte vitrification.

## 4. Materials and Methods

### 4.1. Media and Reagents

Tissue Culture Medium-199 with Hanks’ salts and Tissue Culture Medium-199 with Earle’s salts and Phosphate-buffered saline (PBS) were obtained from Gibco™ Thermo Fisher Scientific (Waltham, MA, USA). All other chemicals not otherwise listed were purchased from Sigma-Aldrich (Diegem, Belgium). All media were filtered before use (0.22 µm filter, Pall Corporation, Ann Arbor, MI, USA).

### 4.2. Collection of Equine Immature Oocytes

Oocyte collection was performed as previously described [7]. Briefly, equine ovaries were obtained from a slaughterhouse and transported in an insulated box to the laboratory at room temperature. Follicles between 5 and 30 mm were aspirated with a vacuum pump (100 mm Hg), scraped with the aspirating needle, and flushed with a prewarmed flushing medium (Equiplus, Minitube, Tiefenbach, Germany). Follicular fluid was collected in sterilized glass bottles and transferred to 100/20 mm Petri dishes. All cumulus-oocyte complexes (COCs) were recovered, washed in Medium 199 with Hank’s salts (Gibco), and pipetted with a 200 µm denudation tip (EZ-tip, Origio, Vreeland, the Netherlands) to remove the outer cumulus cells, leaving the corona radiata. Then, COCs were randomly assigned to vitrification or in vitro maturation. Denuded, partially denuded, and clearly expanded COCs surrounded by a hyaluronan-rich matrix were excluded.

### 4.3. Oocyte Vitrification and Warming

The vitrification and warming process is based on the protocol described by Angel-Velez et al. (2021) [7]. Briefly, oocytes assigned to different vitrification treatments were transferred to 4 mL of base solution (BS), composed of medium 199 with Hank’s salts supplemented with 0.4% (*w*/*v*) bovine serum albumin (BSA) (A6003). Then, four to six oocytes at a time were washed and placed into two droplets of 100 µL of equilibration solution for 25 s. The equilibration solution is composed of BS with 10% (*v*/*v*) ethylene glycol (EG) (#102466) and 10% (*v*/*v*) propylene glycol (PG) (#P4347). Finally, oocytes were transferred to a 100 µL droplet of vitrification solution for 15 s, loaded onto a custom-made minimal volume (<1 µL) cryo-device, and plunged into liquid nitrogen. The time between the placement of oocytes in the vitrification solution and the immersion of the device into the liquid nitrogen was 30–45 s. The vitrification solution was composed of BS with 20% EG, 20% PG, and 0.5 mol/L of trehalose (T0167). Oocytes were loaded using a 200 µm pipette to minimize the volume surrounding the oocytes. For warming, the cryo-device was transferred for 5 min into 4 mL of warming solution 1 (W1), containing BS supplemented with 0.3 mol/L of trehalose. Then, all oocytes were moved to BS until the warming of all oocytes was completed.

### 4.4. In Vitro Maturation

Vitrified-warmed or fresh oocytes were matured in Medium 199 with Earl’s salts (Gibco) containing 10% (*v*/*v*) FBS (Gibco), 50 µg/mL gentamicin, 9.4 µg/mL follicle-stimulating hormone, and 1.88 µg/mL luteinizing hormone (Stimufol, Reprobiol, Ouffet, Belgium). In vitro maturation was performed in groups of 10–30 COCs in 500 µL maturation medium under paraffin oil (Cooper Surgical, Venlo, The Netherlands) at 38.5 °C and 5% CO_2_ in the air. After in vitro maturation or post-warming culture, the oocytes were completely denuded of cumulus using a solution of 0.1% hyaluronidase. Then, oocytes with an extruded polar body were washed in PBS and transferred with a minimum volume to an Eppendorf tube with 2 µL of lysis buffer (5 mM DL-Dithiothreitol (DTT, Molecular Grade, Promega, Madison, WI, USA), 250 units of ribonuclease inhibitor (RNasin^®^ Plus, Promega), and 5.36 M Nonylphenyl-polyethyleneglycol (IGEPAL^®^ CA-630, Sigma-Aldrich)) then snap frozen and stored at −80 °C for RNA extraction. Fresh oocytes and oocytes vitrified/warmed at the immature stage were cultured in in vitro maturation for 28 h and denuded; those with a polar body were kept for two more hours in in vitro maturation before storage. Oocytes vitrified at the MII stage were denuded and vitrified after 28 h of in vitro maturation, warmed, and kept in in vitro maturation for two more hours (post-warming culture) prior to sampling (Figure 1). Hence, all groups had 28 h of in vitro maturation with cumulus cells and 2 h to be denuded.

### 4.5. RNA Extraction and Sequencing

For each group (FR, VMAT, and VIM), total RNA was isolated from five replicates of an average of 23 pooled oocytes (range = 15 to 32) using the RNeasy Micro kit (Qiagen) according to the manufacturer’s protocol. The quality and concentration of the RNA samples were examined using an RNA 6000 Pico Chip (Agilent Technologies, Carlsbad, CA, USA) and a Quant-iT RiboGreen RNA assay kit (Life Technologies, Carlsbad, CA, USA), respectively. After the quality control for total RNA (RNA amount and RIN value), four samples from VIM and VMAT and five from FR were used for further analyses.

Transcriptome library preparation was performed by a QIAseq UPX 3′ transcriptome kit (Qiagen) according to the manufacturer’s instructions with 10 PCR cycles. The quality of the library preparation was checked with a high-sensitivity DNA chip (Agilent Technologies Inc., Santa Clara, CA, USA) and library quantification was performed by qPCR according to the Illumina qPCR quantification protocol (NXTGNT sequencing facility, Ghent, Belgium), followed by the equimolar pooling of libraries based on qPCR. Sequencing was performed on a high throughput Illumina NextSeq 500 flow cell with 20% PhiX spiked in (read 1: 57 cycles; read 2: 27 cycles; and index: 6 cycles).

First, the reads were trimmed with Trim Galore version 0.6.7. (https://github.com/FelixKrueger/TrimGalore, accessed on 7 July 2021) [94], to remove Illumina adapters, poly-A tails, and low-quality bases. The trimmed reads were mapped against the Equus caballus (EquCab3.00, release 108) reference genome using STAR software version 2.7.10b [95]. Unique molecular identifiers (UMIs) were used during the sequencing to characterize the expression levels more accurately and were processed with UMI-tools version 1.1.2 [96]. Finally, RSEM software, version 1.3.1 [97], was used to generate the count tables. The sequenced data were deposited in the National Center for Biotechnology Information (NCBI) Gene Expression Omnibus (GEO) database, (https://www.ncbi.nlm.nih.gov/gds accessed on the 24 February 2023) with accession number GSE225950. One sample from the VMAT group was removed prior to the differential expression analysis because it had deviating quality control metrics (low RIN values, lowest concentration, and few reads).

### 4.6. Differential Gene Expression Analysis

Differential gene expression analysis between the control (FR), VIM, and VMAT group was performed with DESeq2 version 1.32 [98]. Before running DESeq2, only genes that were expressed in at least *n* samples were retained, with *n* equal to the lowest number of samples in a group. This removed lowly expressed genes, which increases the power of the analysis. Afterward, DESeq2 was run with the option ‘independentFiltering = TRUE’ to obtain an additional increase in power. A gene was called differentially expressed (DE) when the Benjamini–Hochberg-adjusted *p*-value was lower than or equal to 0.05 and the absolute value of the log2-fold change was larger than or equal to 1. The heatmaps of the DE genes were computed using the pheatmap package based on the vst-transformed counts of the DESeq2 standard method (median ratio method). Further biological insight was obtained by performing a pathway analysis. Based on a benchmarking study by Geistlinger et al. (2021) [99], pathway analysis with the down-weighting of overlapping genes (PADOG) was chosen as the method to perform GSEA [100]. The method was applied for the gene ontology (GO; http://current.geneontology.org/products/pages/downloads.html, accessed on 30 January 2023) and Kyoto Encyclopedia of Genes and Genomes (KEGG; https://www.genome.jp/kegg/, accessed on 30 January 2023) gene sets. The R package of EnrichmentBrowser 2.28.0 was used to perform both analyses [101]. Pathways with *p* ≤ 0.05 were considered statistically significant. GO terms contain redundant information due to their hierarchical nature. To avoid this redundancy, the initial results of the enrichment analysis were processed further by calculating the semantic similarities between the significant GO terms with the relevance method. Afterward, clustering was performed with the binary cut method. Both steps were implemented in the R package simplifyEnrichment version 1.8.0 [102]. Files with all GO terms in the different comparisons are in Appendix A.

### 4.7. Statistical Analysis for Maturation and Survival Rates

The statistical analyses were performed using R-core (version 4.0.4; R Core Team, Vienna, Austria). The oocyte was considered the unit of interest. Generalized mixed-effects models were used to test the effects of vitrification on the survival rate, in vitro maturation, and efficiency, the replicate was set as a random effect. The differences between groups were assessed using Tukey’s post hoc test. The results of survival, maturation rate, and efficiency are expressed as least square means and standard errors. For the above-mentioned analyses, the R-packages lme4 [103], multcomp [104], and multcompView [105] were utilized. The significance level was set at *p* ≤ 0.05.

## Figures and Tables

**Figure 1 ijms-24-06915-f001:**
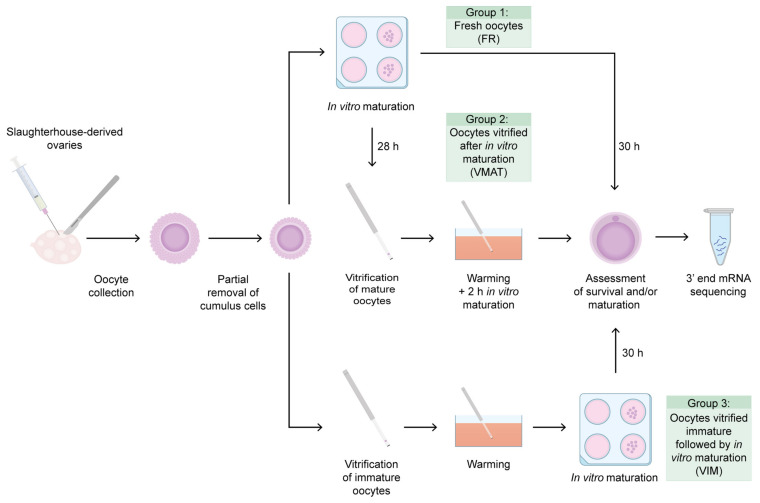
Experimental set-up. After the collection of cumulus-oocyte complexes from slaughterhouse-derived ovaries, the partial removal of cumulus cells was performed. Then, oocytes were assigned to in vitro maturation (n = 422) or to vitrification at the immature stage (n = 204). Oocytes that went directly to in vitro maturation were denuded and those with a visible polar body were designated to continue in vitro maturation for 2 more hours (Group 1: fresh non-vitrified oocytes as a control (FR); n = 111) or to undergo vitrification, warming, and 2 h of post warming culture in in vitro maturation medium (Group 2: oocytes vitrified after in vitro maturation (VMAT); n = 135). Oocytes assigned to vitrification at the immature stage were vitrified, warmed, and in vitro matured for 28 h, denuded, and those with a visible polar body were kept in in vitro maturation for 2 more hours (Group 3: oocytes vitrified immature, warmed, and in vitro matured (VIM); n = 204). The survival rate was assessed after vitrification and warming, with immature oocytes surrounded by the cells of the corona radiata and mature oocytes being denuded. Maturation was assessed by the visualization of an extruded polar body in denuded oocytes. Only mature oocytes with an intact membrane were stored for RNA-seq.

**Figure 2 ijms-24-06915-f002:**
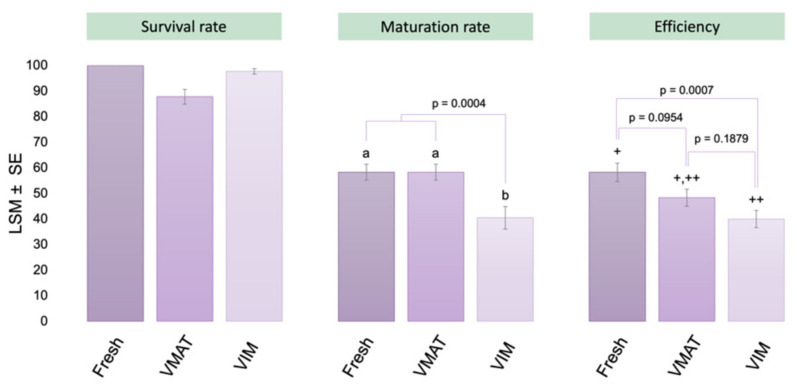
The survival rate is defined as the number of oocytes with an intact oolemma after warming/number of vitrified oocytes. The maturation rate is defined as the number of oocytes with an extruded polar body/number of oocytes exposed to in vitro maturation. The efficiency was calculated as the number of oocytes stored for transcriptomics/initial number of oocytes. Groups with different superscripts (a,b and +,++) differ significantly (*p* < 0.05). The maturation rate in fresh and VMAT is equal since oocytes for those groups were designated after in vitro maturation, and both underwent in vitro maturation together. Group 1: fresh non-vitrified oocytes as a control (FR; n = 190); group 2: oocytes vitrified after in vitro maturation (VMAT; n = 232); group 3: oocytes vitrified before in vitro maturation (VIM; n = 210). Results are presented as n (least square means ± standard error). n = the initial number of oocytes by group.

**Figure 3 ijms-24-06915-f003:**
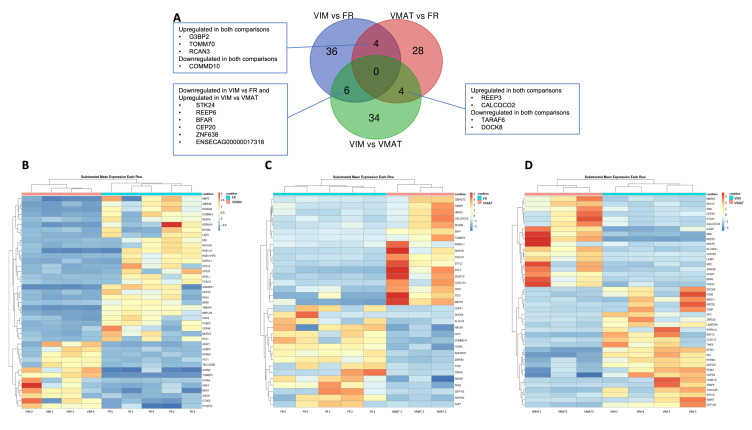
Vitrification at different meiotic stages alters transcriptomics. This figure represents the differentially expressed (DE) genes for three groups: (1) fresh in vitro matured oocytes as a control (FR), (2) oocytes vitrified after in vitro maturation (VMAT), and (3) oocytes vitrified immature, warmed, and in vitro matured (VIM). (**A**) Venn diagram shows the DE genes that overlap between comparisons and the ones specific to each comparison. (**B**) Heatmap generated by the clustering of the DE genes between VIM vs. FR. (**C**) Heatmap generated by the clustering of the DE genes between VMAT vs. FR. (**D**) Heatmap generated by the clustering of the DE genes between VIM vs. VMAT. In the Venn diagram DE categorization (i.e., up or downregulation) is associated with the first factor in the comparison. For all heatmaps, red is upregulation and blue is downregulation, as compared to the mean expression over all samples.

**Figure 4 ijms-24-06915-f004:**
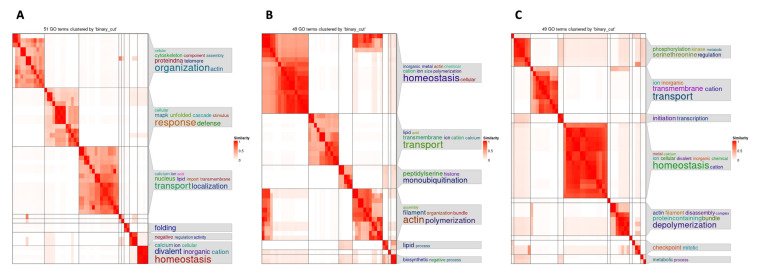
Vitrification affects the pathways involved in oocyte development in vitrified immature (VIM) and metaphase II (VMAT) oocytes, compared to fresh oocytes (FR). This figure represents the SimplifyEnrichment analysis of the resulting GO terms that were clustered by binary cut, enriched, and categorized in biological process for (**A**) VIM vs. FR, (**B**) VMAT vs. FR, and (**C**) VIM vs. VMAT. The enrichment analyses for the cellular components and molecular functions are represented in Appendix A, respectively.

## Data Availability

All data generated or analyzed during this study were included in the manuscript and its Appendix A. Raw data are available from the corresponding author upon reasonable request. The sequenced data were deposited in the National Center for Biotechnology Information (NCBI) Gene Expression Omnibus (GEO) database (https://www.ncbi.nlm.nih.gov/gds, accessed on the 24 February 2023) with accession number GSE225950. In the file, samples named VGV are equal to VIM in the manuscript.

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
