# Peer review of "Transcriptomics Reveal Molecular Differences in Equine Oocytes Vitrified before and after In Vitro Maturation"

_ijms, 2023, doi:10.3390/ijms24086915_

Round 1
Reviewer 1 Report
See attachment

Author Response
Dear
Please found all replies in the attached file.

Reviewer 2 Report
Given the absolute importance of the maternal RNA in the early development of the embryo, the transcriptomic integrity could be a criterion in deciding the quality of oocytes in the zootechnical practice. The authors applied this principle to evaluate the two in vitro maturation protocols being used in the husbandry of horses. To this end, the authors compared the transcriptomes of three groups of oocytes: i) fresh in vitro mature oocytes (FR), ii) oocytes vitrified after in vitro maturation (VMAT), and iii) oocytes vitrified before in vitro maturation (VIM). The three-way comparisons among FR, VMAT and VIM identified a select subset of RNAs being differentially presented after the application of VMAT and VIM protocols, and these gene products were identified as the ones being related to cytoskeletal control and Ca2+ and ion transport activities. The authors also reported that VMAT is slightly advantageous over VIM because it was less deviant from the RNA profile of FR.
It is an interesting study, and the manuscript was well written in a concise way. The updated methodology being used for the study appears to be powerful and impressive. I would make only a couple of critical comments.
[1] The three major pathway groups being differentially presented were RNAs being involved in the controls of cytoskeleton, spindle formation, and calcium and cation ion transport. Although they may represent a fraction of the RNA population, the roles played by them would be critical for the survival and further development of the embryo. Given this physiological importance and the alarmingly diverse results obtained by other studies addressing the same issue, it is recommended that the authors provide the validation data based on the wet bench work such as real-time qPCR of at least some of these differentially regulated genes in the given protocols. Relying entirely on the result of in silico analysis is somewhat insecure, as is often the case.
[2] As a minor point, the abbreviation IVM (in vitro maturation) is very confusing with VIM (vitrified immature, warmed, and in vitro matured). The confusion caused by this similarity in abbreviation actually does a disservice for the readers. For example, in line 284, “vitrification after IVM provides subtle advantages in terms of the mRNA profile of developmentally important genes” perhaps must be specified as “vitrification after IVM (i.e. VMAT) provides subtle advantages in terms of the mRNA profile of developmentally important genes.” What is compared is VIM, but this abbreviation IVM appears so many times in the text and interferes with the narration. Authors many need to find a solution to avoid this confusion.
Author Response

(The authors gave the same response as above.)

Reviewer 3 Report
1. For table 1, kind of confused about the maturation rate for "Fresh" group. The data “243/422” belongs to which group?
2. I also recommend the author showing the data in Table 1 as the plot form, showing p value among different groups and would be more direct for the readers to see the differences.
3. The author should select several identified regulated genes (upregulated as well as downregulated) to do qPCR for further confirmation, which is necessary to make a convicing conclusion based on further analysis.
4. Since the oocyte stage (matured or unmatured) is difficult to clearly distinguish under the COC stage, how did the author separate those into different groups before vitrification (figure 1)? More details need to be clarified in the Method section.
5. More deeply discussion should be addressed based on the limitation of this work.
Author Response

(The authors gave the same response as above.)

Round 2
Reviewer 1 Report
Thank you for addressing the suggested changes
Reviewer 2 Report
In the revised manuscript, the authors have accommodated part of the suggestions raised in my previous report. The authors' explanation of not having performed q-PCR could be understandable, and their commenting on the point in the text might suffice in the given circumstances.
Reviewer 3 Report
Overall the points that I raised have been addressed properly. I agree to publish this paper at the present form.